# Simple, Distributed, and Accelerated Probabilistic Programming

**Dustin Tran**[*]    **Matthew D. Hoffman**[†]    **Dave Moore**[†]    **Christopher Suter**[†]
**Srinivas Vasudevan**[†]    **Alexey Radul**[†]    **Matthew Johnson**[*]    **Rif A. Saurous**[†]

[*]Google Brain, [†]Google

## Abstract

We describe a simple, low-level approach for embedding probabilistic programming in a deep learning ecosystem. In particular, we distill probabilistic programming down to a single abstraction—the random variable. Our lightweight implementation in TensorFlow enables numerous applications: a model-parallel variational auto-encoder (VAE) with 2nd-generation tensor processing units (TPUv2s); a data-parallel autoregressive model (Image Transformer) with TPUv2s; and multi-GPU No-U-Turn Sampler (NUTS). For both a state-of-the-art VAE on 64x64 ImageNet and Image Transformer on 256x256 CelebA-HQ, our approach achieves an optimal linear speedup from 1 to 256 TPUv2 chips. With NUTS, we see a 100x speedup on GPUs over Stan and 37x over PyMC3.[1]

## 1 Introduction

Many developments in deep learning can be interpreted as blurring the line between model and computation. Some have even gone so far as to declare a new paradigm of "differentiable programming," in which the goal is not merely to train a model but to perform general program synthesis.[2] In this view, attention [3] and gating [18] describe boolean logic; skip connections [17] and conditional computation [6, 14] describe control flow; and external memory [12, 15] accesses elements outside a function's internal scope. Learning algorithms are also increasingly dynamic: for example, learning to learn [19], neural architecture search [52], and optimization within a layer [1].

The differentiable programming paradigm encourages modelers to explicitly consider computational expense: one must consider not only a model's statistical properties ("how well does the model capture the true data distribution?"), but its computational, memory, and bandwidth costs ("how efficiently can it train and make predictions?"). This philosophy allows researchers to engineer deep-learning systems that run at the very edge of what modern hardware makes possible.

By contrast, the probabilistic programming community has tended to draw a hard line between model and computation: first, one specifies a probabilistic model as a program; second, one performs an "inference query" to automatically train the model given data [44, 33, 8]. This design choice makes it difficult to implement probabilistic models at truly large scales, where training multi-billion parameter models requires splitting model computation across accelerators and scheduling communication [41]. Recent advances such as Edward [48] have enabled finer control over inference procedures in deep learning (see also [28, 7]). However, they all treat inference as a closed

```python
def model():
    p = ed.Beta(1., 1., name="p")
    x = ed.Bernoulli(probs=p,
                     sample_shape=50,
                     name="x")
    return x
```

**Figure 1:** Beta-Bernoulli program. In eager mode, `model()` generates a binary vector of 50 elements. In graph mode, `model()` returns an op to be evaluated in a TensorFlow session.

```python
import neural_net_negative, neural_net_positive

def variational(x):
    eps = ed.Normal(0., 1., sample_shape=2)
    if eps[0] > 0:
        return neural_net_positive(eps[1], x)
    else:
        return neural_net_negative(eps[1], x)
```

**Figure 2:** Variational program [35], available in eager mode. Python control flow is applicable to generative processes: given a coin flip, the program generates from one of two neural nets. Their outputs can have differing shape (and structure).

system: this makes them difficult to compose with arbitrary computation, and with the broader machine learning ecosystem, such as production platforms [5].

In this paper, we describe a simple approach for embedding probabilistic programming in a deep learning ecosystem; our implementation is in TensorFlow and Python, named Edward2. This lightweight approach offers a low-level modality for flexible modeling—one which deep learners benefit from flexible prototyping with probabilistic primitives, and one which probabilistic modelers benefit from tighter integration with familiar numerical ecosystems.

**Contributions.** We distill the core of probabilistic programming down to a single abstraction—the random variable. Unlike existing languages, there is no abstraction for learning: algorithms may for example be functions taking a model as input (another function) and returning tensors.

This low-level design has two important implications. First, it enables research flexibility: a researcher has freedom to manipulate model computation for training and testing. Second, it enables bigger models using accelerators such as tensor processing units (TPUs) [22]: TPUs require specialized ops in order to distribute computation and memory across a physical network topology.

We illustrate three applications: a model-parallel variational auto-encoder (VAE) [24] with TPUs; a data-parallel autoregressive model (Image Transformer [31]) with TPUs; and multi-GPU No-U-Turn Sampler (NUTS) [21]. For both a state-of-the-art VAE on 64x64 ImageNet and Image Transformer on 256x256 CelebA-HQ, our approach achieves an optimal linear speedup from 1 to 256 TPUv2 chips. With NUTS, we see a 100x speedup on GPUs over Stan [8] and 37x over PyMC3 [39].

## 1.1 Related work

To the best of our knowledge, this work takes a unique design standpoint. Although its lightweight design adds research flexibility, it removes many high-level abstractions which are often desirable for practitioners. In these cases, automated inference in alternative probabilistic programming languages (PPLs) [25, 39] prove useful, so both styles are important for different audiences.

Combining PPLs with deep learning poses many practical challenges; we outline three. First, with the exception of recent works [49, 36, 39, 42, 7, 34], most languages lack support for minibatch training and variational inference, and most lack critical systems features such as numerical stability, automatic differentiation, accelerator support, and vectorization. Second, existing PPLs restrict learning algorithms to be "inference queries", which return conditional or marginal distributions of a program. By blurring the line between model and computation, a lighterweight approach allows any algorithm operating on probability distributions; this enables, e.g., risk minimization and the information bottleneck. Third, it has been an open challenge to scale PPLs to 50+ million parameter models, to multi-machine environments, and with data or model parallelism. To the best of our knowledge, this work is the first to do so.

## 2 Random Variables Are All You Need

We outline probabilistic programs in Edward2. They require only one abstraction: a random variable. We then describe how to perform flexible, low-level manipulations using tracing.

### 2.1 Probabilistic Programs, Variational Programs, and Many More

Edward2 reifies any computable probability distribution as a Python function (program). Typically, the function executes the generative process and returns samples.[3] Inputs to the program—along with any scoped Python variables—represent values the distribution conditions on.

To specify random choices in the program, we use `RandomVariables` from Edward [49], which has similarly been built on by Zhusuan [42] and Probtorch [34]. Random variables provide methods such as `log_prob` and `sample`, wrapping TensorFlow Distributions [10]. Further, Edward random variables augment a computational graph of TensorFlow operations: each random variable x is associated to a sampled tensor $\mathbf{x}^* \sim p(\mathbf{x})$ in the graph.

Figure 1 illustrates a toy example: a Beta-Bernoulli model, $p(\mathbf{x}, \mathbf{p}) = \text{Beta}(\mathbf{p} \,|\, 1, 1) \prod_{n=1}^{50} \text{Bernoulli}(x_n \,|\, \mathbf{p})$, where $\mathbf{p}$ is a latent probability shared across the 50 data points $\mathbf{x} \in \{0, 1\}^{50}$. The random variable x is 50-dimensional, parameterized by the tensor $\mathbf{p}^* \sim p(\mathbf{p})$. As part of TensorFlow, Edward2 supports two execution modes. Eager mode simultaneously places operations onto the computational graph and executes them; here, `model()` calls the generative process and returns a binary vector of $50$ elements. Graph mode separately stages graph-building and execution; here, `model()` returns a deferred TensorFlow vector; one may run a TensorFlow session to fetch the vector.

Importantly, all distributions—regardless of downstream use—are written as probabilistic programs. Figure 2 illustrates an implicit variational program, i.e., a variational distribution which admits sampling but may not have a tractable density. In general, variational programs [35], proposal programs [9], and discriminators in adversarial training [13] are computable probability distributions. If we have a mechanism for manipulating these probabilistic programs, we do not need to introduce any additional abstractions to support powerful inference paradigms. Below we demonstrate this flexibility using a model-parallel VAE.

## 2.2 Example: Model-Parallel VAE with TPUs

Figure 4 implements a model-parallel variational auto-encoder (VAE), which consists of a decoder, prior, and encoder. The decoder generates 16-bit audio (a sequence of $T$ values in $[0, 2^{16} - 1]$ normalized to $[0, 1]$); it employs an autoregressive flow, which for training efficiently parallelizes over sequence length [30]. The prior posits latents representing a coarse 8-bit resolution over $T/2$ steps; it is learnable with a similar architecture. The encoder compresses each sample into the coarse resolution; it is parameterized by a compressing function.

A TPU cluster arranges cores in a toroidal network, where for example, 512 cores may be arranged as a 16x16x2 torus interconnect. To utilize the cluster, the prior and decoder apply distributed autoregressive flows (Figure 3). They split compute across a virtual 4x4 topology in two ways: "across flows", where every 2 flows belong on a different core; and "within a flow", where 4 independent flows apply layers respecting autoregressive ordering (for space, we omit code for splitting within a flow). The encoder splits computation via `compressor`; for space, we also omit it.

The probabilistic programs are concise. They capture recent advances such as autoregressive flows and multi-scale latent variables, and they enable never-before-tried architectures where with 16x16 TPUv2 chips (512 cores), the model can split across 4.1TB memory and utilize up to $10^{16}$ FLOPS. All elements of the VAE—distributions, architectures, and computation placement—are extensible. For training, we use typical TensorFlow ops; we describe how this works next.

## 2.3 Tracing

We defined probabilistic programs as arbitrary Python functions. To enable flexible training, we apply tracing, a classic technique used across probabilistic programming [e.g., 28, 45, 36, 11, 7] as well as automatic differentiation [e.g., 27]. A tracer wraps a subset of the language's primitive operations so that the tracer can intercept control just before those operations are executed.

Figure 5 displays the core implementation: it is 10 lines of code.[4] `trace` is a context manager which, upon entry, pushes a `tracer` callable to a stack, and upon exit, pops `tracer` from the stack. `traceable` is a decorator: it registers functions so that they may be traced according to the stack.

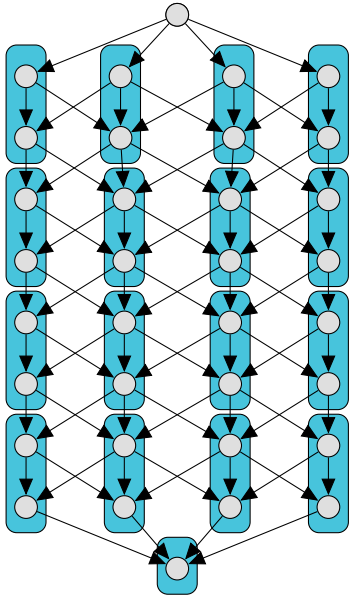

```python
import SplitAutoregressiveFlow, masked_network
tfb = tf.contrib.distributions.bijectors

class DistributedAutoregressiveFlow(tfb.Bijector):
  def __init__(flow_size=[4]*8):
    self.flows = []
    for num_splits in flow_size:
      flow = SplitAutoregressiveFlow(masked_network, num_splits)
      self.flows.append(flow)
    self.flows.append(SplitAutoregressiveFlow(masked_network, 1))
    super(DistributedAutoregressiveFlow, self).__init__()

  def _forward(self, x):
    for l, flow in enumerate(self.flows):
      with tf.device(tf.contrib.tpu.core(l//2)):
        x = flow.forward(x)
    return x

  def _inverse_and_log_det_jacobian(self, y):
    ldj = 0.
    for l, flow in enumerate(self.flows[::-1]):
      with tf.device(tf.contrib.tpu.core(l//2)):
        y, new_ldj = flow.inverse_and_log_det_jacobian(y)
        ldj += new_ldj
    return y, ldj
```

**Figure 3:** Distributed autoregressive flows. **(right)** The default length is 8, each with 4 independent flows. Each flow transforms inputs via layers respecting autoregressive ordering. **(left)** Flows are partitioned across a virtual topology of 4x4 cores (rectangles); each core computes 2 flows and is locally connected; a final core aggregates. The virtual topology aligns with the physical TPU topology: for 4x4 TPUs, it is exact; for 16x16 TPUs, it is duplicated for data parallelism.

```python
import upsample, compressor

def prior():
  """Uniform noise to 8-bit latent, [u1,...,u(T/2)] -> [z1,...,z(T/2)]"""
  dist = ed.Independent(ed.Uniform(low=tf.zeros([batch_size, T/2])))
  return ed.TransformedDistribution(dist, DistributedAutoregressiveFlow(flow_size))

def decoder(z):
  """Uniform noise + latent to 16-bit audio, [u1,...,uT], [z1,...,z(T/2)] -> [x1,...,xT]"""
  dist = ed.Independent(ed.Uniform(low=tf.zeros([batch_size, T])))
  dist = ed.TransformedDistribution(dist, tfb.Affine(shift=upsample(z)))
  return ed.TransformedDistribution(dist, DistributedAutoregressiveFlow(flow_size))

def encoder(x):
  """16-bit audio to 8-bit latent, [x1,...,xT] -> [z1,...,z(T/2)]"""
  loc, log_scale =  tf.split(compressor(x), 2, axis=-1)
  return ed.Normal(loc=loc, scale=tf.exp(log_scale))
```

**Figure 4:** Model-parallel VAE with TPUs, generating 16-bit audio from 8-bit latents. The prior and decoder split computation according to distributed autoregressive flows. The encoder may split computation according to `compressor`; we omit it for space.

```
STACK = [lambda f, *a, **k: f(*a, **k)]

@contextmanager
def trace(tracer):
  STACK.append(tracer)
  yield
  STACK.pop()

def traceable(f):
  def f_wrapped(*a, **k):
    STACK[-1](f, *a, **k)
  return f_wrapped
```

**Figure 5:** Minimal implementation of tracing. `trace` defines a context; any traceable ops executed during it are replaced by calls to `tracer`. `traceable` registers these ops; we register Edward random variables.

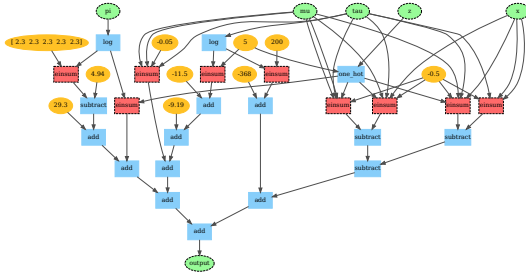

**Figure 6:** A program execution. It is a directed acyclic graph and is traced for various operations such as accumulating log-probabilities or finding conditional independence.

```
def make_log_joint_fn(model):
  def log_joint_fn(**model_kwargs):
    def tracer(rv_call, *args, **kwargs):
      name = kwargs.get("name")
      kwargs["value"] = model_kwargs.get(name)
      rv = rv_call(*args, **kwargs)
      log_probs.append(tf.sum(rv.log_prob(rv)))
      return rv
    log_probs = []
    with trace(tracer):
      model(**model_kwargs)
    return sum(log_probs)
  return log_joint_fn
```

**Figure 7:** A higher-order function which takes a `model` program as input and returns its log-joint density function.

```
def mutilate(model, **do_kwargs):
  def mutilated_model(*args, **kwargs):
    def tracer(rv_call, *args, **kwargs):
      name = kwargs.get("name")
      if name in do_kwargs:
        return do_kwargs[name]
      return rv_call(*args, **kwargs)
    with trace(tracer):
      return model(*args, **kwargs)
  return mutilated_model
```

**Figure 8:** A higher-order function which takes a `model` program as input and returns its causally intervened program. Intervention differs from conditioning: it does not change the sampled value but the distribution.

Edward2 registers random variables: for example, `Normal = traceable(edward1.Normal)`. The tracing implementation is also agnostic to the numerical backend. Appendix A applies Figure 5 to implement Edward2 on top of SciPy.

## 2.4 Tracing Applications

Tracing is a common tool for probabilistic programming. However, in other languages, tracing primarily serves as an implementation detail to enable inference "meta-programming" procedures. In our approach, we promote it to be a user-level technique for flexible computation. We outline two examples; both are difficult to implement without user access to tracing.

Figure 7 illustrates a `make_log_joint` factory function. It takes a `model` program as input and returns its joint density function across a trace. We implement it using a `tracer` which sets random variable `value`s to the input and accumulates its log-probability as a side-effect. Section 3.3 applies `make_log_joint` in a variational inference algorithm.

Figure 8 illustrates causal intervention [32]: it "mutilates" a program by setting random variables indexed by their name to another random variable. Note this effect is propagated to any descendants while leaving non-descendants unaltered: this is possible because Edward2 implicitly traces a dataflow graph over random variables, following a "push" model of evaluation. Other probabilistic operations more naturally follow a "pull" model of evaluation: mean-field variational inference requires evaluating energy terms corresponding to a single factor; we do so by reifying a variational program's trace (e.g., Figure 6) and walking backwards from that factor's node in the trace.

## 3 Examples: Learning with Low-Level Functions

We described probabilistic programs and how to manipulate their computation with low-level tracing functions. Unlike existing PPLs, there is no abstraction for learning. Below we provide examples of how this works and its implications.

```python
import get_channel_embeddings, add_positional_embedding_nd, local_attention_1d

def image_transformer(inputs, hparams):
  x = get_channel_embeddings(3, inputs, hparams.hidden_size)
  x = tf.reshape(x, [-1, 32*32*3, hparams.hidden_size])
  x = tf.pad(x, [[0, 0], [1, 0], [0, 0]])[:, :-1, :]  # shift pixels right
  x = add_positional_embedding_nd(x, max_length=32*32*3+3)
  x = tf.nn.dropout(x, keep_prob=0.7)
  for _ in range(hparams.num_layers):
    y = local_attention_1d(x, hparams, attention_type="local_mask_right",
                           q_padding="LEFT", kv_padding="LEFT")
    x = tf.contrib.layers.layer_norm(tf.nn.dropout(y, keep_prob=0.7) + x, begin_norm_axis=-1)
    y = tf.layers.dense(x, hparams.filter_size, activation=tf.nn.relu)
    y = tf.layers.dense(y, hparams.hidden_size, activation=None)
    x = tf.contrib.layers.layer_norm(tf.nn.dropout(y, keep_prob=0.7) + x, begin_norm_axis=-1)
  logits = tf.layers.dense(x, 256, activation=None)
  return ed.Categorical(logits=logits).log_prob(inputs)

loss = -tf.reduce_sum(image_transformer(inputs, hparams))  # inputs has shape [batch,32,32,3]
train_op = tf.contrib.tpu.CrossShardOptimizer(tf.train.AdamOptimizer()).minimize(loss)
```

**Figure 9:** Data-parallel Image Transformer with TPUs [31]. It is a neural autoregressive model which computes the log-probability of a batch of images with self-attention. Our lightweight design enables representing and training the model as a log-probability function; this is more efficient than the typical representation of programs as a generative process. Embedding and self-attention functions are assumed in the environment; they are available in Tensor2Tensor [50].

## 3.1 Example: Data-Parallel Image Transformer with TPUs

All PPLs have so far focused on a unifying representation of models, typically as a generative process. However, this can be inefficient in practice for certain models. Because our lightweight approach has no required signature for training, it permits alternative model representations.[5]

For example, Figure 9 represents the Image Transformer [31] as a log-probability function. The Image Transformer is a state-of-the-art autoregressive model for image generation, consisting of a Categorical distribution parameterized by a batch of right-shifted images, embeddings, a sequence of alternating self-attention and feedforward layers, and an output layer. The function computes `log_prob` with respect to images and parallelizes over pixel dimensions. Unlike the log-probability, sampling requires programming the autoregressivity in serial, which is inefficient and harder to implement.[6] With the log-probability representation, data parallelism with TPUs is also immediate by cross-sharding the optimizer. The train op can be wrapped in a TF Estimator, or applied with manual TPU ops in order to aggregate training across cores.

## 3.2 Example: No-U-Turn Sampler

Figure 10 demonstrates the core logic behind the No-U-Turn Sampler (NUTS), a Hamiltonian Monte Carlo algorithm which adaptively selects the path length hyperparameter during leapfrog integration. Its implementation uses non-tail recursion, following the pseudo-code in Hoffman and Gelman [21, Alg 6]; both CPUs and GPUs are compatible. See source code for the full implementation; Appendix B also implements a grammar VAE [26] using a data-dependent while loop.

The ability to integrate NUTS requires interoperability with eager mode: NUTS requires Python control flow, as it is difficult to implement recursion natively with TensorFlow ops. (NUTS is not available, e.g., in Edward 1.) However, eager execution has tradeoffs (not unique to our approach). For example, it incurs a non-negligible overhead over graph mode, and it has preliminary support for TPUs. Our lightweight design supports both modes so the user can select either.

```
def nuts(...):
  samples = []
  for _ in range(num_samples):
    state = set_up_trajectory(...)
    depth = 0
    while no_u_turn(state):
      state = extend_trajectory(depth, state)
      depth += 1
    samples.append(state)
  return samples

def extend_trajectory(depth, state):
  if depth == 0:
    state = one_leapfrog_step(state)
  else:
    state = extend_trajectory(depth-1, state)
    if no_u_turn(state):
      state = extend_trajectory(depth-1, state)
  return state
```

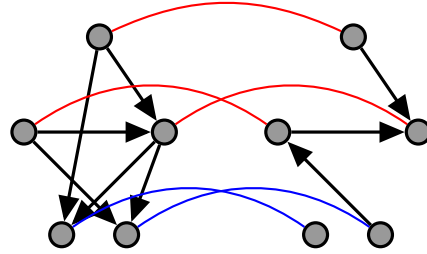

**Figure 11:** Learning often involves matching two execution traces such as a model program's **(left)** and a variational program's **(right)**, or a model program's with data tensors **(bottom)**. Red arrows align prior and variational variables. Blue arrows align observed variables and data; edges from data to variational variables represent amortization.

**Figure 10:** Core logic in No-U-Turn Sampler [21]. This algorithm has data-dependent non-tail recursion.

### 3.3 Example: Alignment of Probabilistic Programs

Learning algorithms often involve manipulating multiple probabilistic programs. For example, a variational inference algorithm takes two programs as input—the model program and variational program—and computes a loss function for optimization. This requires specifying which variables refer to each other in the two programs.

We apply alignment (Figure 11), which is a dictionary of key-value pairs, each from one string (a random variable's `name`) to another (a random variable in the other program). This dictionary provides flexibility over how random variables are aligned, independent of their specifications in each program. For example, this enables ladder VAEs [43] where prior and variational topological orderings are reversed; and VampPriors [46] where prior and variational parameters are shared.

Figure 12 shows variational inference with gradient descent using a fixed preconditioner. It applies `make_log_joint_fn` (Figure 7) and assumes `model` applies a random variable with name `'x'` (such as the VAE in Section 2.2). Note this extends alignment from Edward 1 to dynamic programs [48]: instead of aligning nodes in static graphs at construction-time, it aligns nodes in execution traces at runtime. It also has applications for aligning model and proposal programs in Metropolis-Hastings; model and discriminator programs in adversarial training; and even model programs and data infeeding functions ("programs") in input-output pipelines.

### 3.4 Example: Learning to Learn by Variational Inference by Gradient Descent

A lightweight design is not only advantageous for flexible specification of learning algorithms but flexible composability: here, we demonstrate nested inference via learning to learn. Recall Figure 12 performs variational inference with gradient descent. Figure 13 applies gradient descent on the output of that gradient descent algorithm. It finds the optimal preconditioner [2]. This is possible because learning algorithms are simply compositions of numerical operations; the composition is fully differentiable. This differentiability is not possible with Edward, which manipulates `inference` objects: taking gradients of one is not well-defined.[7] See also Appendix C which illustrates Markov chain Monte Carlo within variational inference.

## 4 Experiments

We introduced a lightweight approach for embedding probabilistic programming in a deep learning ecosystem. Here, we show that such an approach is particularly advantageous for exploiting modern

```
import model, variational, align, x

def train(precond):
  def loss_fn(x):
    qz = variational(x)
    log_joint_fn = make_log_joint_fn(model)
    kwargs = {align[rv.name]: rv
              for rv in toposort(qz)}
    energy = log_joint_fn(x=x, **kwargs)
    entropy = sum([tf.reduce_sum(rv.entropy())
                  for rv in toposort(qz)])
    return -energy - entropy

  grad_fn = tfe.implicit_gradients(loss_fn)
  optimizer = tf.train.AdamOptimizer(0.1)
  for _ in range(500):
    grads = tf.tensordot(precond, grad_fn(x), [[1], [0]])
    optimizer.apply_gradients(grads)
  return loss_fn(x)
```

```
grad_fn = tfe.gradients_function(train)
optimizer = tf.train.AdamOptimizer(0.1)
for _ in range(100):
  optimizer.apply_gradients(grad_fn())
```

**Figure 13:** Learning-to-learn. It finds the optimal preconditioner for `train` (Figure 12) by differentiating the entire learning algorithm with respect to the preconditioner.

**Figure 12:** Variational inference with preconditioned gradient descent. Edward2 offers writing the probabilistic program and performing arbitrary TensorFlow computation for learning.

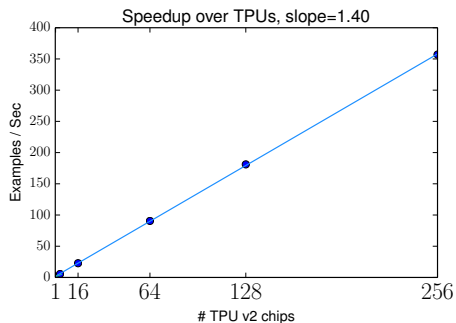

**Figure 14:** Vector-Quantized VAE on 64x64 ImageNet.

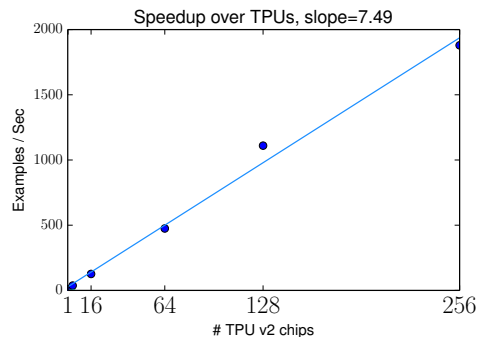

**Figure 15:** Image Transformer on 256x256 CelebA-HQ.

hardware for multi-TPU VAEs and autoregressive models, and multi-GPU NUTS. CPU experiments use a six-core Intel E5-1650 v4, GPU experiments use 1-8 NVIDIA Tesla V100 GPUs, and TPU experiments use 2nd generation chips under a variety of topology arrangements. The TPUv2 chip comprises two cores: each features roughly 22 teraflops on mixed 16/32-bit precision (it is roughly twice the flops of a NVIDIA Tesla P100 GPU on 32-bit precision). In all distributed experiments, we cross-shard the optimizer for data-parallelism: each shard (core) takes a batch size of 1. All numbers are averaged over 5 runs.

## 4.1 High-Quality Image Generation

We evaluate models with near state-of-the-art results ("bits/dim") for non-autoregressive generation on 64x64 ImageNet [29] and autoregressive generation on 256x256 CelebA-HQ [23]. We evaluate wall clock time of the number of examples (data points) processed per second.

For 64x64 ImageNet, we use a vector-quantized variational auto-encoder trained with soft EM [37]. It encodes a 64x64x3 pixel image into a 8x8x10 tensor of latents, with a codebook size of 256 and where each code vector has 512 dimensions. The prior is an Image Transformer [31] with 6 layers of local 1D self-attention. The encoder applies 4 convolutional layers with kernel size 5 and stride 2, 2 residual layers, and a dense layer. The decoder applies the reverse of a dense layer, 2 residual layers, and 4 transposed convolutional layers.

| System | Runtime (ms) |
|---|---|
| Stan (CPU) | 201.0 |
| PyMC3 (CPU) | 74.8 |
| Handwritten TF (CPU) | 66.2 |
| Edward2 (CPU) | 68.4 |
| Handwritten TF (1 GPU) | **9.5** |
| **Edward2 (1 GPU)** | **9.7** |
| **Edward2 (8 GPU)** | **2.3** |

**Table 1:** Time per leapfrog step for No-U-Turn Sampler in Bayesian logistic regression. Edward2 (GPU) achieves a 100x speedup over Stan (CPU) and 37x over PyMC3 (CPU); dynamism is not available in Edward. Edward2 also incurs negligible overhead over handwritten TensorFlow code.

For 256x256 CelebA-HQ, we use a relatively small Image Transformer [31] in order to fit the model in memory. It applies 5 layers of local 1D self-attention with block length of 256, hidden sizes of 128, attention key/value channels of 64, and feedforward layers with a hidden size of 256.

Figure 14 and Figure 15 show that for both models, Edward2 achieves an optimal linear scaling over the number of TPUv2 chips from 1 to 256. In experiments, we also found the larger batch sizes drastically sped up training.

### 4.2 No-U-Turn Sampler

We use the No-U-Turn Sampler (NUTS, [21]) to illustrate the power of dynamic algorithms on accelerators. NUTS implements a variant of Hamiltonian Monte Carlo in which the fixed trajectory length is replaced by a recursive doubling procedure that adapts the length per iteration.

We compare Bayesian logistic regression using NUTS implemented in Stan [8] and in PyMC3 [39] to our eager-mode TensorFlow implementation. The model's log joint density is implemented as "handwritten" TensorFlow code and by a probabilistic program in Edward2; see code in Appendix D. We use the Covertype dataset (581,012 data points, 54 features, outcomes are binarized). Since adaptive sampling may lead NUTS iterations to take wildly different numbers of leapfrog steps, we report the average time per leapfrog step, averaged over 5 full NUTS trajectories (in these experiments, that typically amounted to about a thousand leapfrog steps total).

Table 1 shows that Edward2 (GPU) has up to a 37x speedup over PyMC3 with multi-threaded CPU; it has up to a 100x speedup over Stan, which is single-threaded.[8] In addition, while Edward2 in principle introduces overhead in eager mode due to its tracing mechanism, the speed differential between Edward2 and handwritten TensorFlow code is neligible (smaller than between-run variation). This demonstrates that the power of the PPL formalism comes with negligible overhead.

## 5 Discussion

We described a simple, low-level approach for embedding probabilistic programming in a deep learning ecosystem. For both a state-of-the-art VAE on 64x64 ImageNet and Image Transformer on 256x256 CelebA-HQ, we achieve an optimal linear speedup from 1 to 256 TPUv2 chips. For NUTS, we see up to 100x speedups over other systems.

As current work, we are pushing on this design as a stage for fundamental research in generative models and Bayesian neural networks (e.g., [47, 51, 16]). In addition, our experiments relied on data parallelism to show massive speedups. Recent work has improved distributed programming of neural networks for both model parallelism and parallelism over large inputs such as super-high-resolution images [40]. Combined with this work, we hope to push the limits of giant probabilistic models with over 1 trillion parameters and over 4K resolutions (50 million dimensions).

**Acknowledgements.** We thank the anonymous NIPS reviewers, TensorFlow Eager team, PyMC team, Alex Alemi, Samy Bengio, Josh Dillon, Delesley Hutchins, Dick Lyon, Dougal Maclaurin,

Kevin Murphy, Niki Parmar, Zak Stone, and Ashish Vaswani for their assistance in improving the implementation, the benchmarks, and/or the paper.

## Footnotes

[1]All code, including experiments and more details from code snippets displayed here, is available at http://bit.ly/2JpFipt. Namespaces: `import tensorflow as tf; ed=edward2; tfe=tf.contrib.eager`. Code snippets assume `tensorflow==1.12.0`.

[2]Recent advocates of this trend include Tom Dietterich (https://twitter.com/tdietterich/status/948811925038669824) and Yann LeCun (https://www.facebook.com/yann.lecun/posts/10155003011462143). It is a classic idea in the programming languages field [4].

[3]Instead of sampling, one can also represent a distribution in terms of its density; see Section 3.1.

[4]Rather than implement tracing, one can also reuse the pre-existing one in an autodiff system. However, our purposes require tracing with user control (*tracer* functions above) in order to manipulate computation. This is not presently available in TensorFlow Eager or Autograd [27]—which motivated our implementation.

[5]The Image Transformer provides a performance reason for when density representations may be preferred. Another compelling example are energy-based models $p(x) \propto \exp\{f(x)\}$, where sampling is not even available in closed-form; in contrast, the unnormalized density is.

[6]In principle, one can reify any model in terms of sampling and apply `make_log_joint` to obtain its density. However, `make_log_joint` cannot always be done efficiently in practice, such as in this example. In contrast, the reverse program transformation from density to sampling can be done efficiently: in this example, sampling can at best compute in serial order; therefore it requires no performance optimization.

[7]Unlike Edward, Edward2 can also specify distributions over the learning algorithm.

[8]PyMC3 is actually slower with GPU than CPU; its code frequently communicates between Theano on the GPU and NumPy on the CPU. Stan only used one thread as it leverages multiple threads by running HMC chains in parallel, and it requires double precision.

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
