[Supplementary Material]

## A    Edward2 on SciPy

We illustrate the broad applicability of our tracing implementation by applying SciPy as a back-end.

The implementation wraps `scipy.stats` distributions and registers each `rvs` method as traceable. Variables private from the namescope are explicitly prepended with underscore. Unlike Edward2 on TensorFlow Distributions, generative processes are recorded by calling `rvs` and wrapping Python functions, not Python classes. This is a result of `scipy.stats`'s functional API, which differs from TensorFlow Distributions' object-oriented one.

```python
from scipy import stats

_globals = globals()
for _name in sorted(dir(stats)):
  _candidate = getattr(stats, _name)
  if isinstance(_candidate, (stats._multivariate.multi_rv_generic,
                             stats.rv_continuous,
                             stats.rv_discrete,
                             stats.rv_histogram)):
    _candidate.rvs = traceable(_candidate.rvs)
    _globals[_name] = _candidate
    del _candidate
```

Below is an Edward2 linear regression program on SciPy.

```python
from edward2.scipy import stats as ed  # assuming rvs decorated here

def linear_regression(features):
  coeffs = ed.norm.rvs(loc=0.0, scale=0.1, size=features.shape[1], name="coeffs")
  loc = np.einsum('ij,j->i', features, coeffs)
  labels = ed.norm.rvs(loc=loc, scale=1., size=1, name="labels")
  return labels

log_joint = ed.make_log_joint_fn(linear_regression)

features = np.random.normal(size=[3, 2])
coeffs = np.random.normal(size=[2])
labels = np.random.normal(size=[3])
out = log_joint(features, coeffs=coeffs, labels=labels)
```

See the link to source code for more details.

## B    Grammar Variational Auto-Encoder

Below implements a grammar VAE [26]. It consists of a probabilistic encoder and decoder. It extends probabilistic context-free grammars with neural networks, latent codes, and an encoder for learning representations of discrete structures. The decoder's `logits` is 3-dimensional with shape `[batch_size, max_timesteps, num_production_rules]`.

The encoder takes a string as input and applies `parse_to_one_hot`, a preprocessing step which parses it into a parse tree, extracts production rules from the tree, and converts each production rule into a one-hot vector; it then applies a neural net and outputs a normally-distributed latent code.

The decoder takes a latent code as input and maps it to a sequence of production rules representing the generated string. It applies an RNN followed by a masking step so that the result is a valid sequence of production rules in the grammar. The production rules may then be converted to a string.

```python
import parse_to_one_hot

class ProbabilisticGrammarVariational(tf.keras.Model):
```

```python
    """Amortized variational posterior for a probabilistic grammar."""

    def __init__(self, latent_size):
      """Constructs a variational posterior for a probabilistic grammar."""
      super(ProbabilisticGrammarVariational, self).__init__()
      self.latent_size = latent_size
      self.encoder_net = tf.keras.Sequential([
          tf.keras.layers.Conv1D(64, 3, padding="SAME"),
          tf.keras.layers.BatchNormalization(),
          tf.keras.layers.Activation(tf.nn.elu),
          tf.keras.layers.Conv1D(128, 3, padding="SAME"),
          tf.keras.layers.BatchNormalization(),
          tf.keras.layers.Activation(tf.nn.elu),
          tf.keras.layers.Dropout(0.1),
          tf.keras.layers.GlobalAveragePooling1D(),
          tf.keras.layers.Dense(latent_size * 2, activation=None),
      ])

    def call(self, inputs):
      """Runs the model forward to return a stochastic encoding."""
      net = tf.cast(parse_to_one_hot(inputs), dtype=tf.float32)
      net = self.encoder_net(net)
      return ed.MultivariateNormalDiag(
          loc=net[..., :self.latent_size],
          scale_diag=tf.nn.softplus(net[..., self.latent_size:]),
          name="latent_code_posterior")

class ProbabilisticGrammar(tf.keras.Model):
  """Deep generative model over productions which follow a grammar."""

    def __init__(self, grammar, latent_size, num_units):
      """Constructs a probabilistic grammar."""
      super(ProbabilisticGrammar, self).__init__()
      self.grammar = grammar
      self.latent_size = latent_size
      self.lstm = tf.nn.rnn_cell.LSTMCell(num_units)
      self.output_layer = tf.keras.layers.Dense(len(grammar.production_rules))

    def call(self, inputs):
      """Runs the model forward to generate a sequence of productions."""
      del inputs  # unused
      latent_code = ed.MultivariateNormalDiag(loc=tf.zeros(self.latent_size),
                                              sample_shape=1,
                                              name="latent_code")
      state = self.lstm.zero_state(1, dtype=tf.float32)
      t = 0
      productions = []
      stack = [self.grammar.start_symbol]
      while stack:
        symbol = stack.pop()
        net, state = self.lstm(latent_code, state)
        logits = self.output_layer(net) + self.grammar.mask(symbol)
        production = ed.OneHotCategorical(logits=logits,
                                          name="production_" + str(t))
        _, rhs = self.grammar.production_rules[tf.argmax(production, axis=1)]
        for symbol in rhs:
          if symbol in self.grammar.nonterminal_symbols:
            stack.append(symbol)
        productions.append(production)
        t += 1
      return tf.stack(productions, axis=1)
```

See the link to source code for more details.

## C  Markov chain Monte Carlo within Variational Inference

We demonstrate another level of composability: inference within a probabilistic program. Namely, we apply MCMC to construct a flexible family of distributions for variational inference [38, 20]. We apply a chain of transition kernels specified by NUTS (nuts) in Section 3.2 and the variational inference algorithm specified by train in Figure 12.

```python
import nuts, train

IMAGE_SHAPE = (32, 32, 3, 256)

def model():
  """Generative model of 32x32x3 8-bit images."""
  decoder_net = tf.keras.Sequential([
      tf.keras.layers.Dense(512, activation=tf.nn.relu),
      tf.keras.layers.Dense(np.prod(IMAGE_SHAPE), activation=None),
      tf.keras.layers.Reshape(IMAGE_SHAPE),
  ])

  z = ed.Normal(loc=tf.zeros([FLAGS.batch_size, FLAGS.latent_size]),
                scale=tf.ones([FLAGS.batch_size, FLAGS.latent_size]),
                name="z")
  x = ed.Categorical(logits=decoder_net(z), name="x")
  return x

def variational(x):
  """Variational model given 32x32x3 8-bit images."""
  encoder_net = tf.keras.Sequential([
      tf.keras.layers.Reshape(np.prod(IMAGE_SHAPE)),
      tf.keras.layers.Dense(512, activation=tf.nn.relu),
      tf.keras.layers.Dense(FLAGS.latent_size * 2, activation=None),
  ])

  net = encoder_net(x)
  qz = ed.Normal(loc=net[..., :FLAGS.latent_size],
                 scale=tf.nn.softplus(net[..., FLAGS.latent_size:]),
                 name="qz")
  for _ in range(FLAGS.mcmc_iterations):
    qz = nuts(current_state=qz,
              target_log_prob_fn=lambda z: ed.make_log_joint(model)(x=x, z=z))
  return qz

align_fn = lambda name: {'z': 'qz'}.get(name)
loss = train(0.1)  # uses model, variational, align_fn, x in scope
```

## D  No-U-Turn Sampler

We implement an Edward2 program for Bayesian logistic regression with NUTS.

```python
import build_dataset

def logistic_regression(features):
  """Bayesian logistic regression for labels given features."""
  coeffs = ed.MultivariateNormalDiag(loc=tf.zeros(features.shape[1]), name="coeffs")
  labels = ed.Bernoulli(logits=tf.tensordot(features, coeffs, [[1], [0]]))
  return labels

def make_target_log_prob_fn():
  """Make target density with log-joint function anchored at data."""
  log_joint_fn = ed.make_log_joint_fn(model)
  def target_log_prob_fn(coeffs):
    return log_joint_fn(features=features, coeffs=coeffs, labels=labels)
  return target_log_prob_fn
```

```
features, labels = build_dataset()
coeffs = tf.random_normal(features.shape[1])  # initial state
samples = ed.nuts(current_state=coeffs,
                  target_log_prob_fn=make_target_log_prob_fn())
```

See the link to source code for more details.