[Reviews · NeurIPS 2018]

Reviewer 1



In this submission, the authors describe the design, implementation and performance of Edward2, a low-level probabilistic programming language that seamlessly integrates tensorflow, in particular, tensorflow distribution. The key concept of Edward2 is the random variable, which should be understand as general python functions possibly with random choices in the context of Edward2. Also, continuing the design decision of its first version, Edward2 implements the principle of exposing inference to the users while providing them with enough components and combinators so as to make building custom-inference routines easy. This is different from the principle behind other high-level probabilistic programming systems, which is to hide or automate inference from their users. The submission explains a wide range of benefits of following this principle of exposing inference, such as huge boost in the scalability of inference engines and support for non-standard inference tasks. I support the acceptance of the submission. This submission is about a system, called Edward2. So, it does not have new mathematical theorems or many new ideas. However, the system built is very impressive, and I think that NIPS audience would be very interested to hear about it. The paper gives a good overview on what goes on in Edward2. The explanation on tracing in Edward2 reminds me a lot of poutine in pyro. I think that pyro also heavily rely on tracing, and includes a wide range of library functions for manipulating it. According to my understanding, the implementation of these functions in pyro also uses python context (i.e. a form of aspect-oriented programming) so as to define a new way of running a program or extracting information from program execution. It would help me improve my understanding of Edward2 greatly if there is more thorough comparison between pyro and Edward2. Also, I want to know how Edward2 is different from Edward. One thing that I can easily see is the support for eager evaluation (i.e. no need to wait until the completion of computation graph). What are the other key changes in the high level? * line 245: we are use ===> we are using

Reviewer 2



The paper is solid and describes a major contribution to the field of Statistical Machine Learning. Edward2 is a leap-forward over the existing state-of-the-art, i.e. Edward. I expect it to allow for significantly improved development times for statistical models.

Reviewer 3



The paper describes Edward2, a probabilistic programming language on top of TensorFlow and supports the TPU, GPU and CPU accelerators. It is the successor of the Edward-1 package. Edward-2's main USP is that it has a single abstraction (random variable), and offers no separate abstractions to describe learning itself. The learning procedures are left as functions that are user-defined, making Edward-2 low-level (and powerful -- claim). The paper describes Edward2's tracing mechanism, which allows one to do complex inference procedures, doing interventions on outcomes, etc. The paper does experiments on No-U-Turn sampler, comparing to other packages in the ecosystem. Additionally it does experiments on VAEs, also comparing to other packages in the ecosystem. The results showcase that Edward's usage of GPUs (in NUTS) vs Stan and PyMC3 give it considerable speedups (attributing to the use of GPUs). Additionally for VAEs, they show that their use of TPUs give it a 5x speedup over using GPUs, attributing to the TPU being faster. ------------------------------ The paper is very incomprehensible. I am familiar with TensorFlow and a little bit of probabilistic programs, but I could barely comprehend the paper. I read it 2 and a half times, and I was often short of context. I think the authors seem to have struggled with the page limits and compressed content + removed context. Sections such as 2.2 and 3, 3.1, 3.2 are clearly shortened beyond comprehension for a reader that is not familiar with Edward-1 or whatever context that the authors assume that the reader has. While I have a superficial "tl;dr" of such sections, the point that was being made in the section didn't actually come across. I have very little understanding of what makes Edward2 better. All the code samples are not discussed in sufficient detail either, often "assuming" that the reader "gets" the superiority of what is being presented. While the project seems to be an impressive effort: "PPL with CPU, GPU, TPU support and considerable integration with popular framework such as TensorFlow", the paper should be expanded and probably sent to a venue or journal where one does not have to fight the page limits and can express themselves better. I dont have confidence that as a poster, the authors can do a good job of explaining the framework well. This is my primary reason for rejection of the paper, and I will let the Area Chair make a decision on whether this is a valid reasoning line or not. Lastly, something that seems to be a dishonest "wording" is their sentences around speedups against other packages. Starting from the abstract, and continuing into many places in the paper (8, 44 Table-{1, 2, 3} captions, 235, 243, 9, Table-1 caption, 219, 244) they keep repeating "Edward2 on TPU vs Pyro on GPU" or "Edward2 on GPU vs STAN on CPU". This is not only disingenous, it's dangerous. If you see Table-2, it is clear that Edward2 on TPU vs Edward2 on GPU is a 5x speed gain. So the main speed gain is actually TPU vs GPU, yet they clearly try to mix up the language so that "Edward2" is associated with the speedup rather than "TPU". There's more problematic issues with the benchmarking of TPUs vs GPUs. They compare the TPU to a 1080Ti GPU. They are comparing the latest generation TPU hardware to an older generation GPU hardware. This is not acceptable, especially if they want to make their claim that TPUs are faster. If they compare a TPU against Volta GPUs, will their speedup reduce by a significant amount? Definitely, as Volta's fp32 is a 2x number of flops and has more memory bandwidth. This is my added reason for rejecting the paper in it's current form. Nit: 55, Fig.4, 73 refer Edward2 as "Edward", making it confusing.